# Amlodipine Overdose: Is High Dose Insulin Ready for Prime Time

**Mary Jo S. Farmer** [1,*] **, Anisha Contractor** [1,2] **and Joshua Allgaier** [1]

1    Department of Medicine, UMASS Chan Medical School-Baystate, Springfield, MA 01199, USA
3    Department of Pediatrics, UMASS Chan Medical School-Baystate, Springfield, MA 01199, USA
*    Correspondence: maryjo.farmer@baystatehealth.org

**Abstract:** Overdose of amlodipine, a dihydropyridine calcium channel blocker (CCB), is distinguished from other CCBs due to longer plasma half-life of 30 to 58 h. As current management strategies of CCB overdose are diverse and institution dependent, this retrospective observational study aimed to compare treatment and outcomes data extracted from published case reports of amlodipine overdose with a cohort of patients diagnosed with amlodipine overdose at an urban tertiary medical center. Particular attention was paid to the use of high dose insulin euglycemic therapy (HIET) in treatment of amlodipine overdose. Data was extracted from actual adult patients hospitalized for amlodipine overdose at an urban tertiary medical center up to 2018, and from case reports of amlodipine overdose published between 1997 and 2020. We found a tendency towards earlier and more frequent initiation of HIET over time in management of amlodipine overdose, facilitating hospital discharge. Given the lack of randomized controlled trials comparing vasopressors, HIET, or other therapies, optimal treatment for amlodipine overdose has yet to be definitively established. Based on currently available evidence, a reasonable approach to management of the hemodynamically unstable patient presenting with amlodipine overdose includes vasopressors and inotropes with earlier initiation of HIET.

**Keywords:** amlodipine overdose; amlodipine poisoning; amlodipine toxicity; high dose insulin; calcium channel blocker overdose; dihydropyridine





## 1. Introduction

Calcium channel blockers (CCB) are among the first line medications recommended for the treatment of hypertension with amlodipine being a common choice [1]. Medicare Part-D data from 2015 showed amlodipine as the 4th most prescribed medication with over 38 million claims [2]. Amlodipine is a dihydropyridine CCB with a primary vasodilatory effect on vascular smooth muscle cells via inhibition of calcium influx through L-gated calcium channels in cardiac and vascular smooth muscle cells [3]. Amlodipine overdose typically manifests as vasodilatory shock associated with reflex sinus tachycardia, metabolic acidosis, hyperglycemia, and pulmonary edema [4]. In contrast, non-dihydropyridine CCB medications such as verapamil and diltiazem have more direct effects on cardiac conduction and AV nodal activity so overdose presents as bradycardia and heart block [3].

In the 2019 American Association of Poison Control Centers Toxic Exposure Surveillance System, calcium antagonists were ranked 6th out of the top 25 leading categories of substances associated with fatalities. A listing of pharmaceutical and nonpharmaceutical exposures revealed 1668 (84%) of 2048 fatalities was a pharmaceutical. Of the 1668 pharmaceuticals, 225 were cardiovascular drugs, 66 of which were amlodipine [5].

Calcium channel blocker overdose is a potentially fatal toxicity seen in intensive care units (ICU) across the United States (US), particularly when the patient presents in distributive shock. Management of amlodipine overdose may involve multiple treatment modalities. Despite extensive use of amlodipine, recommendations for treating CCB toxicity in general are primarily based on expert opinion due to low level of evidence [6]. The

evidence sited by the 2017 Expert Consensus Recommendations for the Management of Calcium Channel Blocker Poisoning in Adults [6] is primarily made up of grade D (weak level of evidence) recommendation (https://guides.library.stonybrook.edu/evidence-based-medicine/levels_of_evidence (accessed on 9 December 2022)) and does not differentiate types of CCB overdose.

High dose insulin euglycemic therapy (HIET) has recently become a commonly described therapy for CCB toxicity. The 2017 expert workgroup recommended use of HIET in cases of documented myocardial dysfunction and suggested use of HIET in cases where myocardial dysfunction was not documented because prior case series demonstrated hemodynamic improvement even with dihydropyridine poisoning. Proposed dosing regimens of HIET include a regular insulin bolus of 1 unit/kg followed by an infusion of 1 IU/kg/h with maintenance of euglycemia with dextrose as needed to maintain euglycemia and close monitoring of serum potassium. The workgroup suggested titration up to 10 IU/kg/h only with no respond to first-line therapies as higher dosing is supported by only case series [6,7]. The current dosing recommendations appear to be derived from a 1999 case series of CCB overdose treated with average doses of insulin of 0.5 IU/kg/h [8]. The overall benefits of HIET are thought to outweigh risk of hypoglycemia, hypokalemia or volume overload.

The aims of our study were to use a retrospective literature review to describe trends in hospital course and medical management, in particular HIET, and outcomes of amlodipine overdose and related toxicity including the incidence of death or discharge from the intensive care unit. This information was compared with medical management and outcomes of a recent cohort of patients with amlodipine overdose at a tertiary medical center. The expectation was to identify an increase in HIET use over time and improved outcomes in cases of amlodipine toxicity included in the literature review particularly after publication of the 2017 expert consensus recommendations for management of CCB poisoning in adults. In addition, we describe the medical management of amlodipine toxicity including use of HIET at our tertiary medical center, the role of pharmacy staff input, and whether local care deviated from expert consensus driven care.

## 2. Methods

This project was submitted and approved by the University of Massachusetts Baystate Health Institutional Review Board (IRB). The setting was a 716-bed academic teaching hospital that serves as an urban referral center. Data for this retrospective observational study was collected by two methods: the first being identification of up to 10 eligible patients of any sex, race or ethnicity, age 18 years or greater, hospitalized at our tertiary medical center for amlodipine overdose up to 31 August 2018; the second being identification of case reports of amlodipine overdose published between 1997 and 2020.

### 2.1. Identification of Medical Center Cases

Initial sampling of subjects meeting inclusion criteria was conducted by the institution's Epidemiology/Biostatistics Research Core (EBRC) using the McKesson administrative billing database. Because there is no specific ICD code for amlodipine overdose or toxicity, patients were identified via ICD codes for primary or secondary diagnosis of CCB poisoning (ICD-10 T46.1X1A, T46.1X1B), poisoning by other antihypertensive agents (ICD-9 972.6 and CCB adverse reaction (ICD-10 T46.1X5A) between 1 January 2012 and 31 January 2017. This process identified 171 total patients eligible for further chart review however only 5 patients met requirements for inclusion. Excluded were all cases where amlodipine was not identified as the causative agent in the CCB toxicity or any case with co-ingestion of multiple CCBs.

The protocol was amended to include the ICD codes identified above between 1 January 2012 and 31 August 2018. In addition, the electronic health record system was searched using the Service Line Analytics tool provided by Premier (Charlotte, NC) for the principal diagnosis of: poisoning by beta-adrenoreceptor antagonists, international self-harm, initial encounter (*n* = 8), poisoning by beta-adrenoreceptor antagonists, accidental

(unintentional), initial encounter (*n* = 6); and poisoning by calcium channel blockers, intentional self-harm, initial encounter (*n* = 4). Only 3 of these 18 cases met exclusion criteria. The search was limited to 1 September 2016 through 31 July 2018.

All cases were reviewed by the authors in reverse chronological order. A total of 8 cases with amlodipine related toxicity met inclusion and exclusion criteria using the electronic medical record search criteria described. Data from the 8 cases was extracted by a physician and then confirmed a second time for accuracy by a second physician.

### 2.2. Identification of Case Reports

The initial PubMed search yielded 46 abstracts describing case reports related to amlodipine overdose during 1997 thorough 2019 for which full text articles were requested and data extracted. An updated second PubMed search for case reports related to amlodipine overdose yielded 74 abstracts between 2017 and 2020, 22 for which the full text article was requested for data extraction. PubMed searches were conducted for case reports of amlodipine overdose using the search terms Amlodipine OR amlodipine OR Norvasc AND drug overdose OR overdose OR poison OR toxic OR intoxicant NOT rat OR rats OR mice OR mouse OR murine. All abstracts were reviewed by a physician. Data from the total of 68 case reports was extracted by a physician and then confirmed a second time for accuracy.

### 2.3. Data Extraction

Each of the 68 case reports from the PubMed search and the 8 cases from our own institution was reviewed with the intention of extracting pre-identified variables that were entered in Excel$^{TM}$. Variables included age, gender, identified co-ingestants, Glasgow coma scale (GCS), whether overdose was intentional, initial systolic and diastolic blood pressure, nadir systolic and diastolic blood pressure, initial heart rate, and amount of amlodipine ingested. In cases where only "tachycardia" was reported without a value for heart rate, 110 beats per minute was used.

Treatment information collected included intravenous calcium, glucagon, atropine, vasopressors (norepinephrine, epinephrine, vasopressin, dopamine, ephedrine), inotropes (dobutamine), intravenous fluids, hydrocortisone, decontamination, plasmapheresis, high dose insulin or hyperinsulinemia euglycemia therapy (HIET), dextrose/glucose, cardiac pacing, extracorporeal membrane oxygenation (ECMO) either veno-venous (V-V) or veno-arterial (V-A), lipid emulsion therapy, methylene blue, bicarbonate, isoproterenol, angiotensin II, continuous renal replacement therapy (CRRT) and Molecular Adsorbent Recirculatory System (MARS) [6–9]. The initial dose of insulin and the highest maintenance dose of insulin were reported. Use of renal replacement therapy was recorded as hemodialysis/ultrafiltration (HD/UF), CRRT, sustained low efficiency dialysis (SLED), continuous veno-venous hemofiltration (CVVH), or continuous veno-venous hemodiafiltration (CVVHD).

Therapies administered to patients were collected and delineated between therapies adherent to the 2017 Expert Consensus Recommendations for the Management of Calcium Channel Blocker Poisoning in Adults and those not adherent. Lastly, outcomes data were collected including death, length of stay, ICU admission, intubation and mechanical ventilation, and renal failure. If the case patient was intubated or placed on vasopressors, admission to the ICU was assumed even if not explicitly stated. Laboratory data regarding including lowest glucose (mmol/L), lowest potassium (mmol or mEq/L), lowest sodium (mmol or mEq/L), highest calcium (mg/dL) and highest lactate (mmol/L) were recorded. Other conditions identified included metabolic acidosis, renal failure, cardiac injury, cardiac arrest, conduction delays, first degree heart block, second degree heart block, third degree heart block, bradycardia, hypoxia, pulmonary edema, and acute respiratory distress syndrome (ARDS). Dosage of amlodipine consumed, initial insulin dose and maintenance insulin dose were recorded.

## 3. Results

*3.1. Entire Cohort*

Demographic data is presented in Table 1.

**Table 1.** Demographics, Treatments and Outcomes Data.

| Variable | Number | Cohort Average | Standard Deviation | Number | Case Reports Average | Number | BMC * Average |
|---|---|---|---|---|---|---|---|
| Age (years) | | N = 76 | | | N = 68 | | N = 8 |
| Male | 32 | 50.8 | 16.4 | 26 | 49.3 | 6 | 57.5 |
| Female | 44 | 37.0 | 18.5 | 42 | 36.0 | 2 | 56.0 |
| Initial BP (mmHg) | | | | | | | |
| Systolic | 70 | 82.5 | 27.8 | 62 | 80.9 | 8 | 94.9 |
| Diastolic | 59 | 50.5 | 16.5 | 51 | 48.5 | 8 | 63.4 |
| Nadir BP (mmHg) | | | | | | | |
| Systolic | 48 | 70.5 | 15.9 | 48 | 68.9 | 8 | 80.4 |
| Diastolic | 41 | 43.1 | 13.2 | 41 | 41.1 | 7 | 54.4 |
| Initial HR (bpm) | 61 | 81.2 | 32.3 | 61 | 82.7 | 8 | 69.8 |
| Glasgow Coma Scale | | 13.6 | | | 13.6 | 1 | 13.0 |

| Treatments (2017 Consensus Recommendations) | Entire Cohort | Case Reports | BMC * |
|---|---|---|---|
| Decontamination | 28 (36.8%) | 26 (38.2%) | 2 (25%) |
| IV Ca | 60 (78.9%) | 55 (80.9%) | 5 (62.5%) |
| HIET | 47 (61.8%) | 44 (64.7%) | 3 (37.5%) |
| Norepinephrine | 51 (67.1%) | 47 (69.1%) | 4 (50%) |
| Epinephrine | 30 (39.5%) | 28 (41.2%) | 2 (25%) |
| Dobutamine | 13 (17.1%) | 13 (19.1%) | 0 |
| Atropine | 8 (10.5%) | 7 (10.3%) | 1 (12.5%) |
| Neosynephrine | 12 (15.8%) | 12 (17.6%) | 0 |
| Vasopressin | 19 (25%) | 18 (26.5%) | 1 (12.5%) |
| Dopamine | 30 (39.5%) | 29 (42.6%) | 1 (12.5%) |
| Lipids | 20 (26.3%) | 18 (26.5%) | 2 (25%) |

| Other Treatments | Entire Cohort | Case Reports | BMC * |
|---|---|---|---|
| Ephedrine | 1 (1.3%) | 1 (1.5%) | 0 |
| IV Fluids | 55 (72.4%) | 49 (72.1%) | 6 (75%) |
| Plasmapheresis | 5 (6.6%) | 5 (7.4%) | 0 |
| Hydrocortisone | 4 (5.3%) | 4 (5.9%) | 0 |
| Dextrose/glucose | 33 (43.4%) | 29 (42.6%) | 4 (50%) |
| MARS | 4 (5.3%) | 4 (5.9%) | 0 |
| CRRT | 20 (26.3%) | 19 (27.9%) | 1 (12.5%) |
| Angiotensin II | 1 (1.3%) | 1 (1.5%) | 0 |
| Bicarbonate | 9 (11.8%) | 9 (13.2%) | 0 |
| Isoproterenol | 1 (1.3%) | 1 (1.5%) | 0 |
| Methylene blue | 6 (7.9%) | 6 (8.8%) | 0 |
| Glucagon | 38 (50%) | 35 (51.5%) | 3 (37.5%) |
| ECMO | 7 (9.2%) | 7 (10.3%) | 0 |

Table 1. *Cont.*

| Outcome | Entire Cohort | Case Reports | BMC * |
|---|---|---|---|
| Average length of stay | 11.9 days | 13 days | 5.6 days |
| ICU admission | 58 (76.3%) | 52 (76.5%) | 6 (75%) |
| Endotracheal tube | 48 (63.1%) | 44 (64.7%) | 4 (50%) |
| Death | 12 (15.8%) | 12 (17.6%) | 0 |
| Renal failure | 36 (47.4%) | 32 (47.1%) | 4 (50%) |
| Metabolic acidosis | 24 (31.6%) | 20 (29.4%) | 4 (50%) |
| Cardiac injury | 2 (2.6%) | 2 (2.9%) | 0 |
| Cardiac arrest | 7 (9.2%) | 2 (2.9%) | 0 |
| ROSC | 6 (7.9%) | 6 (8.8%) | 0 |
| Conduction delay | 13 (17.1%) | 11 (16.2%) | 2 (25%) |
| 1st degree block | 4 (5.3%) | 4 (5.9%) | 0 |
| 2nd degree block | 1 (1.3%) | 1 (1.5%) | 0 |
| 3rd degree block | 5 (6.6%) | 5 (7.4%) | 0 |
| Bradycardia | 12 (15.8%) | 9 (13.2%) | 3 (37.5%) |
| Hypoxia | 19 (25%) | 18 (26.5%) | 1 (12.5%) |
| Pulmonary edema | 16 (21.1%) | 16 (23.5%) | 0 |
| ARDS | 2 (2.6%) | 2 (2.9%) | 0 |
| Glucose min (mmol/L) | 6.8 | 7.0 | 4.3 |
| Potassium min (mEq/L) | 3.7 | 4.0 | 3.4 |
| Lactate max (mmol/L) | 7.5 | 9.0 | 3.0 |
| Sodium min (mEq/L) | 133.1 | 132 | 135.7 |
| Calcium max (mg/dL) | 11.9 | 12 | 10.6 |
| Amlodipine dose mean (mg) | 425.6 | 429 | 358.3 |

* BMC = Baystate Medical Center Patients.

Overall, male age ranged from 18 to 92 years. Female age ranged from 14 to 83 years. Average GCS was 13.6. Three patients had a reported GCS of 8 or less.

Average amlodipine dose ingested was 425.6 mg. Co-ingestants to amlodipine were identified in 44/76 (57.9%) cases.

Overdose was intentional in 58/76 (76.3%) cases.

Initial systolic blood pressure (mmHg), initial diastolic blood pressure (mmHg), nadir systolic blood pressure (mmHg), nadir diastolic blood pressure (mmHg), and initial heart rate (bpm) are summarized in Table 1.

Treatment of amlodipine overdose according to 2017 consensus recommendations included decontamination (specified as charcoal, gastric lavage and/or whole bowel irrigation), IV calcium, high dose insulin euglycemic therapy, inotropic support in the form of dobutamine, vasopressor support in the form of norepinephrine, epinephrine, neosynephrine, vasopressin and dopamine, lipids and atropine. Other treatments included ephedrine, intravenous fluids, plasmapheresis, hydrocortisone, dextrose/glucose, MARS, CRRT, angiotensin II, bicarbonate, isoproterenol, methylene blue, glucagon and ECMO. Various types of renal replacement therapies were used including dialysis, ultra-filtration, continuous renal replacement therapy, continuous hemodialysis, or continuous hemodialysis and filtration. There was 1 case where V-V ECMO was used, and 6 cases where V-A ECMO was used. Overall, the most frequent treatments included IV fluids (72.4%), IV calcium (78.9%), norepinephrine (67.1%), HIET (61.8%) and glucagon (50%).

Outcomes. The average length of stay was 11.9 days. Variables collected included ICU admission, endotracheal tube, death, hypoglycemia, electrolyte disturbance, renal failure, metabolic acidosis, cardiac injury, cardiac arrest, ROSC, conduction delay, 1st degree AV block, 2nd degree AV block, 3rd degree AV block, bradycardia, hypoxia, pulmonary edema, ARDS, hyponatremia, hypocalcemia, and lactate level.

Twelve (15.8%) deaths were reported. The majority of the patients were admitted to the medical intensive care unit (76.3%), experienced an endotracheal tube (63.1%) and renal failure (47.4%).

*3.2. Case Reports*

Demographic data is reported in Table 1.

Male age ranged from 18 to 76 years. Female age ranged from 14 to 81 years.

Average GCS was 13.6. Three patients had a reported GCS of 8 or less.

Average dose of amlodipine ingested was 429 mg. Co-ingestants to amlodipine were identified in 36/68 (52.9%) cases.

Overdose was intentional in 52/68 (76.5%) cases.

Initial systolic blood pressure (mmHg), initial diastolic blood pressure (mmHg), nadir systolic blood pressure (mmHg), nadir diastolic blood pressure (mmHg), and initial heart rate (bpm) are summarized in Table 1.

Treatment of amlodipine overdose according to 2017 consensus recommendations included decontamination (specified as charcoal, gastric lavage and/or whole bowel irrigation), IV calcium, high dose insulin euglycemic therapy, inotropic support in the form of dobutamine, vasopressor support in the form of norepinephrine, epinephrine, neosynephrine, vasopressin and dopamine, lipids and atropine. Other treatments included ephedrine, intravenous fluids, plasmapheresis, hydrocortisone, dextrose/glucose, MARS, CRRT, angiotensin II, bicarbonate, isoproterenol, methylene blue, glucagon and ECMO. Various types of renal replacement therapies were used including dialysis, ultra-filtration, continuous renal replacement therapy, continuous hemodialysis, or continuous hemodialysis and filtration. There was 1 case where V-V ECMO was used, and 6 cases where V-A ECMO. The most frequent treatments included IV calcium (80.9%), IV fluids (72.1%), norepinephrine (69.1%), HIET (64.7%) and glucagon (51.5%).

Outcomes. The average length of stay was 13 days. Variables collected included ICU admission, endotracheal tube, death, hypoglycemia, electrolyte disturbance, renal failure, metabolic acidosis, cardiac injury, cardiac arrest, ROSC, conduction delay, 1st degree AV block, 2nd degree AV block, 3rd degree AV block, bradycardia, hypoxia, pulmonary edema, ARDS, hyponatremia, hypocalcemia, and lactate level.

Twelve (17.6%) deaths were reported. The majority of the patients were admitted to the medical intensive care unit (76.5%), experienced an endotracheal tube (64.7%) and renal failure (47.1%).

*3.3. BMC Cohort*

Demographic data is reported in Table 1.

Male age ranged from 42 to 92 years. Female age ranged from 29 to 83 years.

Average GCS was 13.6.

Average dose of amlodipine ingested was 358.3 mg. Co-ingestants to amlodipine were identified in 8/8 (100%) cases.

Overdose was intentional in 6/8 (75%) cases.

Initial systolic blood pressure (mmHg), initial diastolic blood pressure (mmHg), nadir systolic blood pressure (mmHg), nadir diastolic blood pressure (mmHg), and initial heart rate (bpm) are summarized in Table 1.

Treatment of amlodipine overdose according to 2017 consensus recommendations included decontamination (specified as charcoal, gastric lavage and/or whole bowel irrigation), IV calcium, high dose insulin euglycemic therapy, vasopressor support in the form of norepinephrine, epinephrine, vasopressin and dopamine, lipids and atropine. Other treatments included intravenous fluids, dextrose/glucose, CRRT and glucagon. Hemodialysis and ultrafiltration were the types of renal replacement therapies used. The most frequent treatments included IV fluids (75%), IV calcium (62.5%) and dextrose/glucose (50%).

Outcomes. The average length of stay was 5.6 days. Variables collected included ICU admission, endotracheal tube, death, hypoglycemia, electrolyte disturbance, renal failure, metabolic acidosis, cardiac injury, cardiac arrest, ROSC, conduction delay, 1st degree AV block, 2nd degree AV block, 3rd degree AV block, bradycardia, hypoxia, pulmonary edema, ARDS, hyponatremia, hypocalcemia, and lactate level.

No deaths were reported. The majority of the patients were admitted to the medical intensive care unit (75%), experienced an endotracheal tube (50%), renal failure (50%) and metabolic acidosis (50%).

## 4. HIET

Use of insulin in the combined cohort of case report and actual patient data increased over time as shown in Figure 1a,b. Both the initial doses of insulin and maintenance doses of insulin increased over time. Six patients in whom insulin was not part of the amlodipine overdose treatment plan died, whereas 5 patients who received insulin as part of their amlodipine overdose treatment plan died.

Interestingly, there were no deaths amongst our 8-patient cohort. Insulin was part of the amlodipine overdose treatment plan in cases occurring in 2016 and 2017 (total of 4 cases). Insulin was not part of the treatment plan in cases occurring in 2012 (2 cases), 2013 (1 case) and 2015 (2 cases).

(**a**)

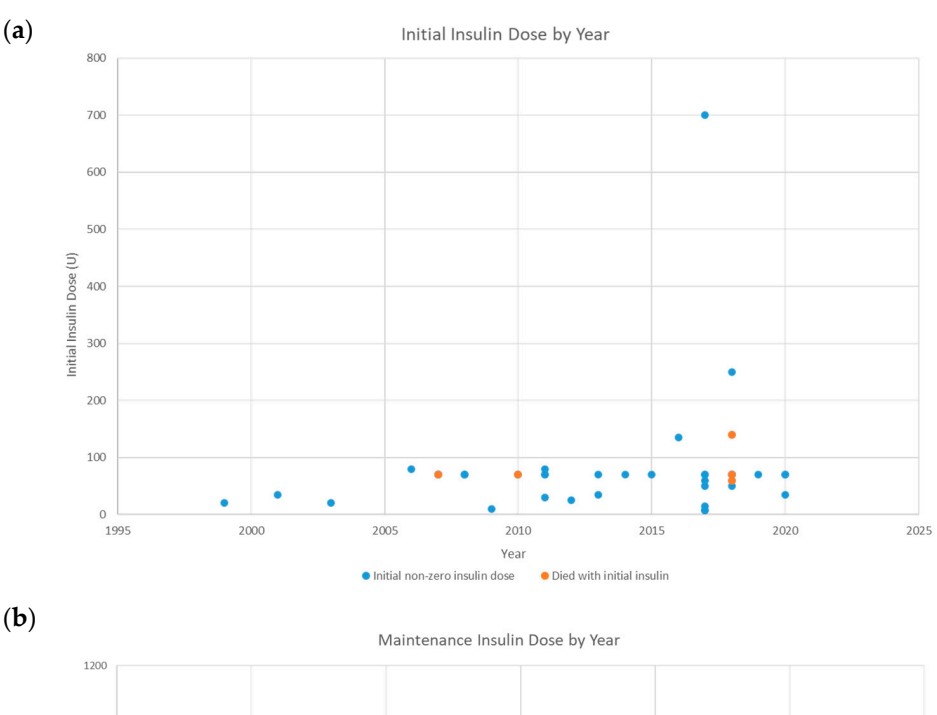

(**b**)

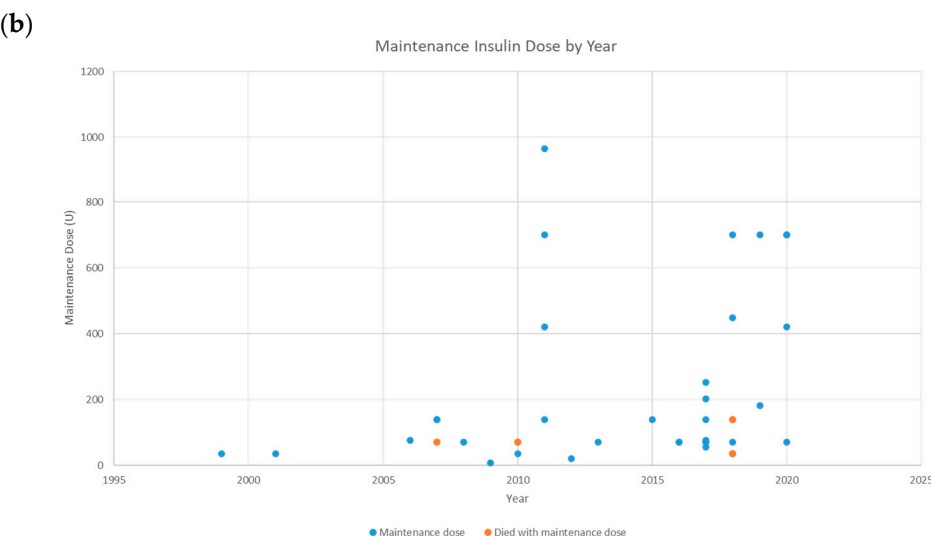

**Figure 1.** Regular Insulin Dosing. (**a**). Initial insulin dose by year (**b**). Maintenance insulin dose by year.

## 5. Discussion

Amlodipine is prescribed as the besylate salt in tablets of 2.5, 5 and 10 mg with the usual initial antihypertensive oral dose being 2.5 mg once daily in geriatric and debilitated

patients to 5.0 mg once daily in adult patients, and the maximum dose of 10 mg once daily (www.ncbi.nlm.nih.gov/books/NBK519508 (accessed on 9 December 2022)). Unlike other calcium channel blockers, amlodipine has a very low metabolic clearance permitting once a day dosing to maintain a near-constant plasma concentration. The therapeutic plasma level of amlodipine ranges from 5 to 18 mg/L. Amlodipine has the longest plasma half-life of 30–50 h compared to all calcium channel blockers [10,11], as well as a large volume of distribution (21 L/kg), strong binding to albumin (90–95%), and a relative lack of negative inotropy [12]. Hypotension and other signs of amlodipine toxicity may appear or last up to 7 days due to the long drug half-life, making it clinically prudent to monitor patients following amlodipine overdose for 24–36 h [13].

In 1998 Adams [14] reported the first documented survivor of a massive amlodipine overdose, although the patient later expired due to complications. In 2019, 66 fatalities were attributed to amlodipine [5]. How best to treat patients who present in an emergent clinical state from a CCB overdose and more specifically amlodipine overdose has yet to be definitely established. Currently, there are no randomized control trials comparing vasopressors, HIET, or any other therapy and there are a number of experimental therapies currently being evaluated [15]. The 2017 recommendations propose and algorithm where the following first-line measures are strongly recommended in symptomatic CCB poisoning: IV calcium, with norepinephrine or epinephrine in the presence of shock, and high-dose IV insulin in the presence of myocardial dysfunction. Insulin monotherapy in the presence of cardiac dysfunction, atropine in symptomatic bradycardia, and dobutamine or epinephrine in the presence of cardiogenic shock are additional lower strength recommendations. For CCB toxicity refractory to first-line treatments, incremental doses of high-dose insulin therapy up to 10 units/kg/h in the presence of myocardial dysfunction, pacemaker for unstable bradycardia or high-grade AV block without cardiac dysfunction and IV lipid emulsion therapy are suggested. In cases with severe refractory shock and peri-arrest, high-dose insulin therapy up to 10 units/kg/h even in the absence of myocardial dysfunction and veno-arterial extracorporeal membrane oxygenation (V-A ECMO) are suggested in addition to pacemaker and IV lipid emulsion therapy. Treatment of patients who present up to 1 h following ingestion of a potentially toxic amount of CCB includes observation and consideration of decontamination [6,7]. A multimodal therapeutic approach is often used according to specifics of the presenting situation [16].

The clinical manifestation of CCB toxicity include profound hypotension with bradycardia and/or conduction blocks with non-dihydropyridine CCB overdoses such as Verapamil and Diltiazem, due to primarily myocardial effects. These patients can deteriorate to cardiogenic shock rapidly. Dihydropyridine CCBs such as amlodipine, nicardipine and nifedipine act primarily on vascular smooth muscle and cause hypotension and possible reflex tachycardia or normal heart rate however some of the selectivity may be lost with significant overdoses [17]. The loss of pharmacological selectivity at high serum amlodipine levels was described by Ebihara [16].

Boyer [18] described that blockade of L-type calcium channels in myocardial cells, vascular smooth muscle cells and beta islet cells preventing the intracellular influx of calcium result in the clinical features of CCB toxicity. Antagonism of these channels produces 4 cardiovascular effects: negative chronotropy or bradycardia via sinoatrial node inhibition, negative inotropy or decreased cardiac contractility, negative dromotropy or conduction delay thorough inhibition of AV node, and peripheral arterial vasodilation, as well as other systemic effects including hypoinsulinemia, hyperglycemia, metabolic acidosis, and shock. Calcium channel blockers inhibit L-type calcium channels in pancreatic islet cells, reducing insulin secretion which results in hyperglycemia and reduced cardiac glucose utilization [18].

In an unstressed state, myocardial cells and smooth muscle cells oxidize free fatty acids for metabolic energy but during the state of shock myocytes shift to metabolism of glucose for energy [11]. Hypoinsulinemia may prevent the uptake of glucose by myocytes, precipitating the shock state by promoting a loss of inotropy and reduced peripheral

vascular resistance, resulting in decreased mechanical efficiency of the heart thereby further contributing to a vicious cycle of poor tissue perfusion and acidosis [11].

Although the exact mechanism of action of hyperinsulinemia-euglycemia therapy is not well understood, HIET is known to improve inotropy, improve peripheral vascular resistance and reverse acidosis by facilitating uptake of carbohydrates by myocytes and smooth muscle cells [18,19].

In addition to the 2017 Expert Consensus Recommendations [6], the HIET regimens reported to the literature vary from 0.1 to 10 units/kg/h for the rate of the insulin infusion [16]. While evidence for definitive dosing is lacking, the literature promotes use of regular insulin 1 unit/kg administered with dextrose 0.5 g/kg for an initial bolus dose. Recommended maintenance doses are insulin 0.5 units/kg/h increased to 2 units/kg/h if no improvement is seen within 30 min of administration and dextrose 0.5 g/kg/h [17].

Chudow 2017 [20] commented that there is no universal treatment algorithm in the setting of acute amlodipine overdose particularly when selectivity is lost and negative cardiac effects are displayed. Accepted treatment options supported in the literature are found in Table 2 and summarized by St. Onge et al., 2017 [6].

**Table 2.** Treatment Options for Amlodipine Overdose.

| **Basic Resuscitation—ABC, IV Access, Supplemental Oxygen, Monitor, EKG** |
| --- |
| GI decontamination including gastric lavage within 1–2 h and administration of activated charcoal |
| Central line |
| Calcium—30 mL bolus of 10% calcium gluconate for injection followed by infusion at 10 mL/h |
| Glucagon—5–10 mg IV bolus up to 15 mg of injection glucagon followed by infusion at 3–5 mg/h |
| Vasopressors and crystalloids |
| Inotropes |
| High dose insulin euglycemic therapy (HIET) 1 unit/kg regular human insulin IV bolus followed by an infusion at 0.5–1 units/kg/h. |
| Lipid emulsion therapy |
| Molecular adsorbent recirculating system (MARS) |
| Extra corporal membrane oxygenation (ECMO) |
| Intra-aortic balloon pump |

Remember: Atropine is unlikely to improve bradycardia in severe overdoses because these patients often have infra nodal blocks. Similarly, pacing may not improve cardiac output due to decrease inotropy.

In addition to the above measures, Akarca [21] included cardiac pacing and introduced the concept of intravenous lipid emulsion therapy (ILE) in CCB intoxication to be used as a late salvage option when other measures have failed In addition, therapeutic plasma exchange (TPE) for acute drug overdose has a Category II status in the guidelines of the American Association of Blood Banks (AABB) and the American Society for Apheresis (ASFA) meaning TPE is generally accepted as adjunctive therapy [22].

We found the use of HIET in 4 of the amlodipine overdoses treated at our medical center resulted in recovery from the overdose and hospital discharge. Similarly, the majority of case series patients were discharged when insulin therapy was part of the treatment plan. These outcomes speak to the importance of early initiation of HIET in cases of amlodipine intoxication. Administration of HIET is ideally done in collaboration with hospital pharmacy staff with use of an approved protocol. The medical team engaged with pharmacy staff during and independent of ICU rounds to discuss HIET dosing, blood glucose values, and ensuring availability of insulin at high doses for prolonged periods of administration.

## 6. Limitations

Limitations of this study include its retrospective nature, and the small number of actual patients presenting with amlodipine overdose. In many of the case reports and actual patients, amlodipine was not the only agent ingested in toxic amounts. Although this scenario parallels clinical practice and complicates treatment, the clinician is forced to rely on bystander history along with presenting signs and symptoms when developing a plan of care. Ideally, a randomized clinical trial controlled for first line therapies could substantiate the benefit of HIET in amlodipine overdose seen in our experience.

## 7. Conclusions

Optimal treatment for amlodipine overdose has yet to be established. Currently, there are no randomized control trials comparing vasopressors, HIET, or other therapies. The question to be dealt with is how to treat these patients who present in an emergent clinical state. In addition, further study is needed to better understand the role and timing of HIET in treatment of amlodipine overdose so as to maximize benefit.

Based on the evidence currently available, a reasonable approach to CCB overdose is to use pressors and inotropes with or without HIET as the foundation in management of the hemodynamically unstable patient. High dose insulin euglycemic therapy can be quite effective when begun early. Important adjuncts to remember are glucagon to reverse bradycardia despite the short effect, and calcium.

If all else fails, mechanical rescue therapy in the form of ECMO or an intra-aortic balloon pump should be tried. Extra corporal membrane oxygenation can support the patient in cardiovascular collapse until all the drug is metabolized out of the body. Cardiothoracic surgery consultation or transfer to a hospital with ECMO capability might be needed. If ECMO is not an option, an intra-aortic balloon pump should be placed with the intent of supporting the patient until the drug is metabolized.

**Author Contributions:** Conceptualization, J.A. and M.J.S.F.; methodology, J.A. and M.J.S.F.; software, J.A., A.C. and M.J.S.F.; validation, J.A., A.C. and M.J.S.F.; formal analysis, J.A., A.C. and M.J.S.F.; investigation, J.A., A.C. and M.J.S.F.; resources, J.A. and M.J.S.F.; data curation, M.J.S.F., J.A. and A.C.; writing—original draft preparation, J.A., A.C. and M.J.S.F.; writing—review and editing, J.A., A.C. and M.J.S.F.; visualization, M.J.S.F., J.A. and A.C.; supervision, M.J.S.F.; project administration, J.A. and M.J.S.F. All authors have read and agreed to the published version of the manuscript.

**Funding:** This research received no external funding.

**Institutional Review Board Statement:** The study was conducted in accordance with the Declaration of Helsinki, and approved by the Institutional Review Board (or Ethics Committee) of University of Massachusetts Baystate Health (Protocol 1163042-4; 29 May 2018).

**Informed Consent Statement:** Not applicable.

**Data Availability Statement:** The data presented in this study are available on request from the corresponding author.

**Acknowledgments:** Adam Pesaturo, Pharmacy Coordinator, Medication Formulary, Baystate Medical Center, Springfield, MA. Peter St Marie, Data Scientist, Epidemiology and Biostatistics Research Core, Baystate Medical Center, Springfield, MA.

**Conflicts of Interest:** The authors declare no conflict of interest.

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
