# Peer review of "Amlodipine Overdose: Is High Dose Insulin Ready for Prime Time"

_hearts, doi:10.3390/hearts4010001_

Round 1
Reviewer 1 Report
Amlodipine, a long-acting dihydropyridine derivative, is one of the most frequently used antihypertensive drugs, especially in ederly patients. These patients often require multidrug therapy and polypharmacy increases the risk of drug-to-drug interactions. In this light presented study is very important and shed a light on management of amlodypine poisoning. Results show that average glucose level was 6,8 mmol/L in entire investigated cohort, which can be interpret as euglycemia. Metabolic acidosis was revealed in over 30% of patients. Since insulin may improve carbohydrate metabolism in cardiac myocytes, HIET is justified in patients who overdosed amlodypine. However, I was a bit confused with presented data. I suggest to combine all tables into one to follow the results more easily. Also, abbreviation BMC cohort should be explained directly. If possible cite doi: 10.3390/ijms23010286 in dissusion section.
Author Response
We thank Reviewer 1 for the thoughtful comments. The glucose level reported for each case and actual patient was the lowest value reported given a concern for hypoglycemia, particularly in cases where HIET was used. Alternatively, many of the references document that hyperglycemia is present upon initial presentation of patient with amlodipine overdose and persists even when HIET is initiated. For this reason, the authors are willing to go back to the cases and patient data sets and extract the highest dose of serum glucose reported and add a new data field called glucose maximum.
Once this point is clarified, the authors will then combine the tables adding the glucose maximum field, creating one large data table.
Dr. Farmer will attempt to integrate the information found in the given DOI into the manuscript.
The term BMC has been clarified by use of a footnote or endnote after the current table.
Please see attached as well for further clarifications made in the manuscript.

Reviewer 2 Report
The manuscript entitled "Amlodipine overdose: is high dose insulin ready for prime time" is well designed and well written. Please take attention revise and unify tables 1, and 2 before publication.
Author Response
We thank reviewer 2 for the thoughtful comments. The authors agree Tables 1 & 2 will be unified and merged before publication, however we are awaiting further clarification from Reviewer 1 before proceeding with this task.
I noticed that Reviewer 2 checked that "methods could be improved". Does reviewer 2 have any suggestions as to how this could be accomplished?
attached response letter.
